

# Diel variations in planktonic ciliate community structure in the northern South China Sea and tropical Western Pacific

Chaofeng Wang[1,2,3], Yi Dong[1,2,3], Michel Denis[4], Li Zhao[1,2,3], Haibo Li[1,2,3], Shan Zheng[1,5], Wuchang Zhang[1,2,3] , Tian Xiao[1,2,3]

[1]CAS Key Laboratory of Marine Ecology and Environmental Sciences, Institute of Oceanology, Chinese Academy of Sciences, Qingdao 266071, China

[2]Laboratory for Marine Ecology and Environmental Science, Qingdao National Laboratory for Marine Science and Technology, Qingdao 266237, China

[3]Center for Ocean Mega-Science, Chinese Academy of Sciences, Qingdao 266071, China

[4]Aix Marseille Université, Université de Toulon, CNRS/INSU, IRD, Institut Méditerranéen d'Océanologie (MIO), Marseille cedex09 13288, France

[5]Jiaozhou Bay Marine Ecosystem Research Station, Institute of Oceanology, Chinese Academy of Sciences, Qingdao 266071, China

**Correspondence:** Wuchang Zhang (wuchangzhang@qdio.ac.cn)

**Abstract:**

Though diel variations are geographically widespread phenomena among phytoplankton and zooplankton, knowledge is limited regarding diel variations in planktonic ciliate (microzooplankton) community structure. In this study, we analyzed diel variations in community structure of planktonic ciliates in the northern South China Sea (nSCS) and tropical Western Pacific (tWP). Hydrological characteristics during day and night were slightly different over both the nSCS and tWP, while ciliate average abundance in night was clearly higher than in day in the upper 200 m. The differences in weighted mean depth (WMD) for aloricate ciliates and tintinnids revealed that they preformed diel vertical migrations. In the nSCS, the WMDs of both aloricate ciliates and tintinnids were higher in day than in night. However, in the tWP, the WMDs of aloricate ciliates were higher in day than in night, whereas it was the opposite for tintinnids whose WMDs were lower in day than in night, indicating that they were in phase opposition with aloricate ciliates. In both the nSCS and tWP, abundance proportions of large size-fraction aloricate ciliates in night were higher than in day. While for tintinnids, abundance proportion of large lorica oral diameter in night were lower than in day. The relationship between environmental factors and ciliate abundance pointed out that depth and temperature were main factors influencing



aloricate ciliate and tintinnid abundances in both day and night. For some dominant
tintinnid species, chlorophyll *a* was another important factor influencing diel vertical
distribution. Our results provide fundamental data for better understanding the diel
vertical migration ecological role of planktonic ciliates in the tropical Western Pacific
Ocean.
**Key words:** Planktonic ciliate; diel variation; community structure; northern South
China Sea; tropical Western Pacific

**1 Introduction**
Planktonic ciliates taxonomically belong to phylum Ciliophora, class Spirotrichea,
subclass Oligotrichia and Choreotrichia (Lynn 2008), and they morphologically consist
of tintinnids and aloricate ciliates. Marine planktonic ciliates are important components
of microzooplankton as primary consumers of pico- (0.2–2 μm) and nano-(2–20 μm)
sized plankton, and important food items of metazoans and fish larvae (Stoecker et al.,
1987; Dolan et al., 1999; Gómez 2007). Therefore, they play an important role in
material circulation and energy flow from the microbial food web into the classical food
chain (Azam et al., 1983; Pierce and Turner, 1992; Calbet and Saiz, 2005). Owing to
their rapid growth rates and sensitivity to environmental changes, ciliates have been
considered as effective bioindicators in different water masses (Kim et al., 2012; Wang
et al., 2021a, 2022a).
Diel variations, which are common phenomenon in marine plankton, include
variations in abundance, behavior, physiology, feeding and cell-division (e.g., Haney
1988; Vaulot and Marie, 1999; Hays et al., 1998, 2001; Anna et al., 2020). The diel
behavior of phytoplankton was found to be affected by light-dependence of cell growth
and continuous losses to grazing in the tropical and subtropical seas (Vaulot et al., 1995;
Vaulot and Marie, 1999; Binder and DuRand, 2002; Li et al., 2022), which eventually
led to community diel variations. For example, in the northern South China Sea (nSCS)
at night, the abundance and cell size of picophytoplankton (*Prochlorococcus*,
*Synechococcus*, and picoeukaryotes) were respectively higher and smaller than during
the day (Li et al., 2022). With respect to marine planktonic zooplankton, most studies





dedicated to meso-/macro-zooplankton, have established that they often perform diel
vertical migration (descending at dawn and ascending in late afternoon and evening),
and have higher abundance in night than in day (e.g., Ohman 1990; Ringelberg 1999;
Tarling et al., 2002; Cohen and Forward, 2005a, 2005b; Ringelberg 2010; Liu et al.,

2020).

In contrast, studies related to planktonic ciliate (microzooplankton) diel variations

remain limited, even though several investigations on planktonic-ciliate diel variations
were conducted in different habitats (Dale 1987; Stocker et al., 1989; Passow 1991;
Suzuki and Taniguchi, 1997; Olli 1999; Pérez et al., 2000; Rossberg and Wickham,
2008). In oceanic waters, the autotrophic ciliate *Mesodinium rubrum* was shown to
migrate from subsurface to surface waters at daytime in the Baltic Sea (Passow 1991;
Olli 1999). Some micro-sized heterotrophic ciliates appeared to migrate from a depth
of 20-30 m (day) to the surface (night) in the northwestern Mediterranean Sea (Pérez et
al., 2000). But in the shelf and slope waters of the Georges Bank (northwest Atlantic)
(Stocker et al., 1989), and the Toyama Bay (Japan Sea) (Suzuki and Taniguchi, 1997),
abundance of planktonic ciliates varied little during the day and night, suggesting that
they may not migrate vertically. In the eutrophic shallow waters of a Germany gravel
pit lake characterised by stable water stratification, Rossberg and Wickham (2008)
found that the abundances of several dominant ciliate species were significantly higher
in day than in night. We found no study on ciliate diel variations in tropical oceanic
waters.

The South China Sea is the largest semi-enclosed basin in the western Pacific

Ocean (Su, 2004), and the tropical Western Pacific (tWP) holds the largest warm pool
area with sea-surface temperature > 28 ℃ throughout the year (Cravatte et al., 2009).
Many studies were conducted on ciliate communities in the northern slope of the South
China Sea (Feng et al., 2013; Liu et al., 2016; Wang et al., 2019, 2021a; Sun et al., 2021)
and the tWP (Gómez 2007; Sohrin et al., 2010; Kim et al., 2012; Wang et al., 2020,
2021b). However, none of these studies addressed ciliate community diel variations,
nor provided any comparison between the nSCS and tWP.

In the present study, we hypothesized that planktonic ciliate community structure



might differ between day and night and that ciliates performed diel vertical migrations.
By examining time-series data of ciliate community structure in the nSCS and tWP, we
aimed to determine diel variations in: (1) ciliate abundance and biomass at each
sampled depth; (2) overall abundance and abundance proportions of different size-
fractions of aloricate ciliates; (3) tintinnid composition and the abundance proportions
of different lorica oral diameter (LOD) size-classes. The output of this study is expected
to be of great help in monitoring microzooplankton diel vertical migration and
forecasting their ecological influence in the marginal and tropical oceanic seas.

**2 Materials and methods**
**2.1 Study area and sample collection**

The variation of ciliate vertical distribution was addressed by conducting two

time-series sampling in the upper 500 m at two distinct sites, Station (St.) S1 in nSCS
and St. P1 in tWP, during two different cruises (Fig. 1). St. S1 was visited from 29 to
31 March 2017 aboard R.V. "Nanfeng", and St. P1 from 2 to 3 June 2019 aboard R.V.
"Kexue". During 48 h (St. S1) or 24 h (St. P1) sampling periods, seawater samples were
collected by using a CTD (Sea-Bird Electronics, Bellevue, WA, USA) - rosette carrying
12 Niskin bottles of 12 L each (Table 1). In the nSCS, the sampling depths were 3, 10,
25, 50, DCM (deep Chl a maximum layer), 100, 200 and 500 m; in the tWP, the
sampling depths were 3, 30, 50, 75, DCM, 150, 200, 300 and 500 m. Casts were
approximately launched every 6 h, the CTD determining vertical profiles of
temperature, salinity and chlorophyll *a in vivo* fluorescence (Chl *a*). A total of 117
seawater samples were collected for planktonic ciliate community structure analysis.
For each depth, 1 L seawater sample was fixed with acid Lugol's (1% final
concentration) and stored in darkness at 4 ℃ during the cruise.

**2.2 Sample analysis and species identification**

In the laboratory, water samples were concentrated to approximately 200 mL by

siphoning off the supernatant after the sample had settled for 60 h. This settling and
siphoning process was repeated until a final concentrated volume of 50 mL was



125 achieved, which was then settled in two Uterm öhl counting chambers (25 mL per

126 chamber) (Uterm öhl 1958) for at least 24 h. Planktonic ciliates were counted using an

127 Olympus IX 73 inverted microscope (100× or 400×) according to the process of

128 Uterm öhl (1958) and Lund et al. (1958).

129  For each species, size (length, width, according to shape) of the cell (aloricate

130 ciliate) or lorica (tintinnid, especially length and oral diameter) were determined for at

131 least 10 individuals if possible. Aloricate ciliates were categorized into small (10-20

132 μm), medium (20-30 μm) and large (>30 μm) size-fractions for maximum body length

133 of each individual following Wang et al. (2020). Tintinnid taxa were identified

134 according to the size and shape of loricae following Kofoid and Campbell (1929, 1939),

135 Lynn (2008), Zhang et al. (2012) and Wang et al. (2019, 2021a, 2021b). Tintinnid

136 species richness in each station was highlighted by the number of tintinnid species that

137 appeared in that station. Because mechanical and chemical disturbance during

138 collection and fixation can detach the tintinnid protoplasm from the loricae (Paranjape

139 and Gold, 1982; Alder 1999), we included empty tintinnid loricae in cell counts.

141 **2.3 Data processing**

142  Ciliate volumes were estimated using appropriate geometric shapes (cone, ball,

143 and cylinder). Tintinnid carbon biomass was estimated using the equation:

144    $C = V_i \times 0.053 + 444.5$ (Verity and Lagdon, 1984)

145  Where $C$ (μg C $L^{-1}$) is the carbon biomass, $V_i$ (μm$^3$) is the lorica volume. We used

146 a conversion factor of carbon biomass for aloricate ciliates of 0.19 pg/μm$^3$ (Putt and

147 Stoecker, 1989). Calculation of ciliate water column average abundance and biomass

148 was following Yu et al. (2014) and Wang at al. (2022b). We used the Margalef index

149 ($d_{Ma}$) (Margalef 1958) and Shannon index ($H'$) (Shannon 1948) to test tintinnid

150 diversity indices in day and night variations. Biogeographically, classification of

151 tintinnid genera (Cosmopolitan, species distributed widespread in the world ocean;

152 Warm Water, species observed in both coastal systems and open waters throughout the

153 world ocean, but absent from sub-polar and polar waters) was based on Pierce and

154 Turner (1993) and Dolan and Pierce (2013). The weighted mean depth (WMD) was



used to reflect ciliate community diel vertical migration, and was calculated using the
formula:

WMD = SUM(a$_i$d$_i$) / SUM(a$_i$) (Frost and Bollens, 1992)

Where a$_i$ is the abundance at depth d$_i$, and d$_i$ is the midpoint of each sampling
layer.
The dominance index (Y) of tintinnid species in one assemblage was calculated
using formula:

$Y = \frac{N_i}{(N \times f_i)}$ (Xu and Chen, 1989)

Where $N_i$ is the number of individuals of species $i$ in all samples, $f_i$ is the
occurrence frequency of species $i$ in all samples and $N$ is the total number of species.
Species with $Y \geq 0.02$ represented the dominant species in an assemblage.
Distributional data of sampling stations, ciliates and environmental parameters
(Depth, temperature, salinity, and Chl $a$) were visualized by ODV (Ocean Data View,
Version 5.0, Reiner Schlitzer, Alfred Wegener Institute, Bremerhaven, Germany),
Surfer (Version 13.0, Golden Software Inc., Golden, CO, United States), OriginPro
2021 (Version 9.6, OriginLab Corp., United States), and Grapher (Version 12.0, Golden
Software Inc., Golden, CO, United States). Correlation analysis between environmental
and biological variables was performed using SPSS (Version 16, SPSS Inc., IBM Corp.,
Armonk, NY, USA).

**3 Results**
**3.1 Hydrology and ciliate vertical distribution**
Hydrological characteristics throughout day and night were slightly different in
the nSCS and tWP (Fig. 2). Temperature decreased with depth from surface (3 m) to
500 m. However from surface to 100 m depth at nSCS, its average values at each depth
in daytime were slightly higher by 0.20 ± 0.16 ℃ than in night. In contrast, in the tWP,
the average temperature values at each depth in daytime were slightly lower by 0.24 ±
0.26 ℃ than in night (Fig. 2). Salinity first increased from surface to approximately 150
m, then decreased to 500 m in both the nSCS and the tWP. Salinity average values at



depths from surface to 100 m in daytime at both the nSCS and the tWP, were slightly
higher by 0.01 ±0.01 and 0.01 ±0.03, respectively, than in night. At each depth, salinity
in the nSCS was higher than in tWP (Fig. 2). Chlorophyll *a* in vivo fluorescence (Chl
*a*) showed similar characteristics in both day and night, while average deep Chl *a*
maximum (DCM) layers in day of the nSCS (82.5 ±6.5 m) and the tWP (101.5 ±12.0
m) were deeper than in night (nSCS: 77.0 ±4.5 m; tWP: 98.7 ±5.1 m), respectively
(Fig. 2).

High ciliate abundance (≥ 200 ind. L$^{-1}$) and biomass (≥ 0.5 µg C L$^{-1}$) values were

mainly observed in the nSCS upper 100 m and the tWP upper 150 m, and then decreased
down to 500 m depth (Fig. 2). Aloricate ciliates were a dominant group in both the
nSCS and tWP (Fig. S1). The vertical profiles of ciliate average abundance and biomass
showed bimodal (in the surface and DCM layers) patterns throughout day and night in
both the nSCS and tWP. However, there were some differences in details (Fig. 2). From
surface to 200 m depth, average abundance and biomass of ciliates in night were higher
than in day in both the nSCS and tWP. But highest values in the nSCS were in surface
layers, whereas in the tWP, they were in the DCM layers (Fig. 2). At surface layers of
the nSCS, average abundance (517.0 ±132.6 ind. L$^{-1}$) and biomass (2.8 ±2.0 µg C L$^{-1}$)
in night were 1.3 and 2.0 folds higher than in day (413.3 ±77.6 ind. L$^{-1}$ and 1.4 ±0.7
µg C L$^{-1}$), respectively. At DCM layers of the tWP, average abundance (476.7 ±21.4
ind. L$^{-1}$) and biomass (1.3 ±0.2 µg C L$^{-1}$) in night were 1.4 and 1.1 folds higher than in
day (347.0 ±103.2 ind. L$^{-1}$ and 1.2 ±0.9 µg C L$^{-1}$), respectively (Fig. 2). There were
almost no differences between day and night in waters deeper than 200 m in the nSCS
and tWP, respectively (Fig. 2; Fig. S1).

The weighted mean depth (WMD) of ciliate community during day and night

differed in both the nSCS and tWP. With respect to total ciliates and aloricate ciliates,
average WMD in day were slightly higher than in night in both the nSCS and tWP.
When it came to tintinnids, average WMD in day of the nSCS (61.1 ±8.5 m) was 3.4
m higher than in night (57.7 ±3.6 m), but this value in day (76.7 ±8.3 m) of the tWP
was 16.6 m lower than in night (93.3 ±8.7 m). Average WMDs of aloricate ciliates and
tintinnids in the tWP were deeper than in the nSCS in both day and night (Fig. 3).

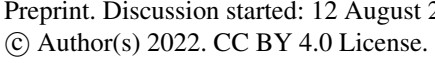




### 3.2 Water column average abundance and biomass of ciliates

Average integrated abundance and biomass of ciliate varied significantly during
day and night in both the nSCS and tWP (Fig. 4). In the nSCS, the night water-column
average-abundance of total ciliates (136.3 $\pm$7.7 ind. L$^{-1}$), aloricate ciliates (126.2 $\pm$8.2
ind. L$^{-1}$) and tintinnids (10.1 $\pm$2.0 ind. L$^{-1}$) were higher than in day (116.1 $\pm$8.1 ind. L$^-$
$^1$, 106.3 $\pm$7.3 ind. L$^{-1}$ and 9.8 $\pm$1.2 ind. L$^{-1}$), respectively. In contrast, the night water-
column average-biomass of tintinnids (0.017 $\pm$0.003 µg C L$^{-1}$) was lower than in day
(0.020 $\pm$0.004 µg C L$^{-1}$). In the tWP, the night water-column average abundances and
biomasses of total ciliates, aloricate ciliates and tintinnids were higher than in day (Fig.
4). As to night and day variations, the water-column average abundances and biomasses
of total ciliates and aloricate ciliates in night and day were higher in the tWP than in the
nSCS (Fig. 4). Although the night and day water-column average-abundance of
tintinnids was higher in the tWP than in the nSCS, their night and day water-column
average-biomass values were lower in the tWP than in the nSCS (Fig. 4).

### 3.3 Aloricate ciliate size-fractions

In the nSCS and tWP, day and night average abundances and abundance
proportions of each aloricate ciliate size-fraction, were different (Fig. S2). Generally,
in the upper 150 m of both the nSCS and tWP, average abundance of small (10-20 µm),
medium (20-30 µm) and large (>30 µm) size-fractions of aloricate ciliate were higher
in night than in day. In contrast, in the nSCS upper 150 m and tWP upper 75 m,
abundance proportions of the small size-fraction were lower in night than in day. As to
day and night variations in approximately the upper 80 m of nSCS and tWP, the average
abundance and abundance proportion of the large size-fraction of aloricate ciliates were
higher in the nSCS than in the tWP. However, the opposite was observed at 100 m (Fig.
S2). There was almost no difference between day and night abundances and abundance
proportions in waters deeper than 200 m in both the nSCS and tWP (Fig. S2).



### 3.4 Tintinnid assemblage

### 3.4.1 Tintinnid abundance, composition, and diversity index

In total, 69 tintinnid species from 27 genera were identified through the study (Table S1). Among them, 57 tintinnid species from 23 genera and 51 tintinnid species from 25 genera were observed at the nSCS and tWP stations, respectively. Tintinnid abundance ranged from 0 - 87 ind. $L^{-1}$ and 0 - 73 ind. $L^{-1}$ in the nSCS and tWP, respectively. Both high abundance ($\geq$ 10 ind. $L^{-1}$) and species richness ($\geq$ 5) occurred in the upper 200 m (Fig. 5). In the nSCS, Margalef ($d_{Ma}$) and Shannon ($H'$) indices were higher in night than in day. However, in the tWP, these diversity indices hardly varied from day to night (Fig. 5). As for tintinnid biogeography type, cosmopolitan and warm water genera were the dominant groups at both sites. Regarding diel variations in both the nSCS and tWP, more cosmopolitan and warm water species were found in night than in day (Table S2).

### 3.4.2 Vertical distribution of dominant species

Five and eight dominant species ($Y \geq 0.02$) occurred in the nSCS and tWP, respectively. Among them, only *Salpingella faurei* and *Proplectella perpusilla* appeared in both sites (Table S1). As for dominant species in the nSCS, *S. faurei* and *Epiplocylis acuminata* exhibited a higher abundance at 50 m and DCM layer in night than in day. In contrast, in the surface layer, *Dadayiella ganymedes* and *Steenstrupiella steenstrupii* were present in higher abundance at day than at night. The *P. perpusilla* abundance was higher at 25 m and 50 m depths in night than in day, but at DCM and 100 m depths, its abundances were lower in night than in day (Fig. 6; Fig. S3).

In the tWP, the abundance of *S. faurei*, *P. perpusilla*, *Ascampbelliella armilla, Acanthostomella minutissima* and *Metacylis sanyahensis* at DCM layer was clearly higher in night than in day. The surface-layer abundance of *Canthariella brevis* and *Protorhabdonella curta* was obviously higher during the day than at night. The *Eutintinnus hasleae* abundance was higher in night than in day from 50 to 200 m (except DCM) (Fig. 6; Fig. S3).





### 3.4.3 Lorica oral diameter size-classes, lorica length and abundance proportion of tintinnid species

Number of species richness and high average abundance in tintinnid LOD (lorica oral diameter) size-classes were consistent throughout the day and night in both the nSCS and tWP, but there were some slight differences (Fig. 7). Species richness in night over the nSCS (49) and tWP (45) were slightly higher than in day (nSCS: 44, tWP: 44), respectively (Fig. 7; Table S1). Highest species richness and average abundance were in the 28-32 μm LOD size-class during the day and night in both the nSCS and tWP. Between day and night, the second highest species richness in the nSCS and tWP were 32-36 μm and 24-28 μm LOD size-class, respectively. While the second highest average abundance were 12-16 μm and 20-24 μm LOD size-class, respectively. Generally, average abundance of most tintinnid LOD size-classes were higher in night than in day. However, these night and day values were similar in the tWP (Fig. 7).

In the nSCS, abundance proportion of *S. faurei* (highest, 16.8%) and *D. ganymedes* (second highest, 15.7%) were lower in day than in night (18.4% and 16.1%, respectively). Abundance proportion of *S. steenstrupii* (third highest, 9.3%) was higher in day than in night (5.5%). In the tWP, *S. faurei* (9.2%), *C. brevis* (8.8%) and *P. curta* (7.1%) had the three highest abundance proportion in day. In night, however, species with the three highest abundance proportion changed to *A. minutissima* (9.5%), *S. faurei* (9.0%) and *P. perpusilla* (6.2%) (Fig. 7). Additionally, tintinnid species with lorica length greater than 150 μm had higher abundance proportion in day than in night in both the nSCS and tWP (Fig. 7).

### 3.5 Relationship between ciliate abundance and environmental factors

Temperature-salinity-plankton diagrams showed that aloricate ciliate size-fractions (small, medium, and large) and tintinnid dominant species behaved within different temperature and salinity ranges that varied from day and night in the nSCS and tWP (Fig. 8). For aloricate ciliates, temperature and salinity range of small (10-20 μm), medium (20-30 μm), and large (> 30 μm) size-fractions were wider in night than in day in both the nSCS and tWP (Fig. 8; Fig. S4). Regarding differences between the

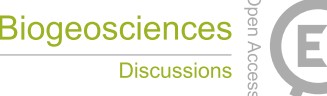

two sites, the temperature range of each aloricate ciliate size-fraction with abundance >
100 ind. L$^{-1}$ in the nSCS (23.1-24.8 °C, average 24.3 ±0.5 °C) was lower than that in the
tWP (24.8-29.8 °C, average 27.8 ± 1.9 °C). As for tintinnids, all dominant species
(except *D. ganymedes*) in the nSCS had temperature ranges wider in night than in day,
and their higher abundance was associated with salinity higher in day (except *E.*
*acuminata*) than in night (Fig. 8). In the tWP, all dominant species (except *S. faurei*)
corresponded to wider salinity ranges in night than in day (Fig. 8; Fig. S5).
Relationships between ciliate abundances and environmental factors (depth,
temperature, salinity, and Chl *a*) during day and night, differed in both the nSCS and
tWP (Table 2). In the nSCS and tWP, Aloricate ciliates and total ciliates had strong
significant negative and positive correlations with depth and temperature, respectively,
whether in day time or at night. As for dominant tintinnids in the nSCS, *S. faurei* had
significant positive correlation with Chl *a* in night, but no correlation with Chl *a* in
daytime. *P. perpusilla* had significant positive correlation with Chl *a* in day, but no
correlation with Chl *a* in night (Table 2). In the tWP, *S. faurei*, *P. perpusilla*, *M.*
*sanyahensis* and total tintinnids were not correlated with Chl *a* in day, but they exhibited
significant correlations in night (Table 2).

**4 Discussion**
**4.1 Diel vertical distribution variations of ciliate community**
In oceans, zooplankton have evolved diverse strategies of survival, and diel
vertical migration is the key to understanding the functioning of ciliates in marine
planktonic microbial food web ecosystems (Ringelberg 2010; Bandara et al., 2021).
There are two historical perspectives regarding ciliate diel-vertical-migration
(excluding the phototrophic ciliate *Mesodinium rubrum*, which exhibits obvious diel
vertical migration) (Lindholm and Mörk, 1990; Passow 1991; Olli 1999). Stocker et al.
(1989), and Suzuki and Taniguchi (1997), found that most planktonic ciliates do not
show perceivable vertical migration. In contrast, other studies provided evidence that
ciliates indeed perform diel vertical migration (Dale 1987; Pérez et al., 2000; Rossberg
and Wickham, 2008). The weighted mean depth (WMD) was used to test diel vertical

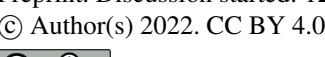



migration in copepods (Frost and Bollens, 1992). To our knowledge, we are the first to
propose using the WMD for testing ciliate diel-vertical-migration. We found that the
WMD of aloricate ciliates and tintinnids deferred between day and night in both the
nSCS and tWP (Fig. 3), supporting our hypothesis that they do perform diel vertical
migration.
The ciliate vertical distribution patterns were the same between day and night in
both the nSCS and tWP, with abundance peaks in surface and DCM layers, respectively.
These results were similar to previous ones established in the western Pacific Ocean
(Yang et al., 2004; Sohrin et al., 2010; Wang et al., 2019, 2020, 2021b). However, the
studies that previously investigated the ciliate vertical distribution, did not assess
potential differences between day and night vertical distribution. Therefore, our study
provides more accurate data on ciliate diel-vertical-migration in the nSCS and tWP.
Additionally, our results in the upper 200 m provide evidence that ciliate abundance
and biomass were higher in night than in day in both the nSCS and tWP (Fig. 2).
Zooplankton distribution in waters mainly depends on phytoplankton presence (Daro
1988; Ursella et al., 2018). Thus, it is possible that the availability of more food items
(flagellates, picoeukaryotes, *Prochlorococcus*, *Synechococcus* and heterotrophic
bacteria) in night than in day explains the higher ciliate abundance in night (Olli 1999;
Oubelkheir and Sciandra, 2008; Li et al., 2022).

**4.2 Diel variations in aloricate ciliate size-fractions**
Abundance proportions of different aloricate ciliate size-fractions have rarely
been reported in the nSCS and tWP. In the tropical Pacific Ocean, average abundance
proportions of small size-fraction (10-20 μm) of aloricate ciliates to total ciliates ranged
from 38 to 50% (from surface to 200 m depth), and it belonged to the dominant group
at each depth in most stations (Yang et al., 2004; Wang et al., 2020, 2021b). Our results
for the small size-fraction of aloricate ciliates in the tWP are consistent with those of
previous studies in both day and night. In the upper 100 m of both nSCS and tWP sites,
the large size-fraction (>30 μm) of aloricate ciliates had more pronounced diel
variations than those of the small size-fraction (Fig. S2). We speculated that the large





size-fraction of aloricate ciliates were migrating along distances longer than those
crossed by the small size-fraction. This phenomenon may be similar to that observed in
meso-/macro-zooplankton in the nSCS (Liu et al., 2020), equatorial Pacific Ocean
(Roman et al., 2002), subtropical and subarctic North Pacific Ocean (Steinberg et al.,
2008), and northwest Mediterranean (Isla et al., 2015).

**4.3 Potential reason for tintinnid diel variations**
The LOD of a tintinnid is closely related to the size of its preferred food item
(approximately 25% of the LOD) (Dolan 2010). Our results showed that tintinnid
abundance was higher in night than in day, while biomass decreased in both the nSCS
and tWP (Figs. 3 and 7). We also found that abundance and abundance proportion of
the 12-16 μm LOD size-class of tintinnids was higher in night than in day. These results
suggest that both LOD size-classes of tintinnids and the size of their preferred food
items were smaller in night than in day. The night-dominant smaller cell sizes of food
items (picoeukaryotes, *Prochlorococcus*, *Synechococcus*) in night than in day (Li et al.,
2022) may be coupled with the observed tintinnid diel variations.
For photosynthetic organisms, cell division generally occurs at night and/or in the
late afternoon (Jacquet et al., 2001; Binder and DuRand, 2002), which eventually leads
to higher abundance in night than in day (Li et al., 2022). Heterotrophic tintinnids feed
on prey picoplankton and heterotrophic bacteria in the ocean. Our study showed that
the night tintinnid abundance was higher than in day for two possible reasons: 1)
oceanic tintinnid species have stronger cell division in midnight than in day in tropical
Pacific waters (Heinbokel 1987); and 2) predation on picoplankton and heterotrophic
bacteria occurred primarily at night (Tsai et al., 2005; Ribalet et al., 2015; Connell et
al., 2020). Further studies on growth rates and cell division of tintinnid species are
needed to better characterizing their diel vertical migration in the Pacific Ocean.

**4.4 Differences of ciliate community between the nSCS and tWP oceanic waters**
Vertical distribution patterns of planktonic ciliates were of bimodal type with
abundance peaks at surface and DCM layers in both the nSCS and tWP, but highest



abundances occurred in surface and DCM layers of the nSCS and tWP, respectively
(Fig. 2). Our results are consistent with Wang et al. (2019), who discovered this
phenomenon and proposed a hypothesis to verify it. The nSCS is located at the
convergence area of the shelf and slope waters where exchanges often occur with
nutrient loaded waters originating from the Pearl River through surface current (e.g.,
Cheung et al., 2003; Huang et al., 2003; Liu et al., 2010; Shu et al., 2018). In contrast,
the tWP is located at a tropical Pacific warm pool surrounded year-round by
oligotrophic oceanic water. This may be the main reason for the surface layer ciliate
abundance in the nSCS clearly higher than in tWP.
Aloricate ciliates were dominant groups at each sampled depth of both sites (Fig.
S1), which was similar to previous observations in adjacent seas (Yang et al., 2004;
Gómez, 2007; Sohrin et al., 2010; Wang et al., 2019, 2021a, 2021b). As for tintinnid
assemblages, we identified more species in the nSCS (57 species) than in the tWP (51
species) (Table S1), which was not consistent with previous investigations (Li et al.,
2018; Wang et al., 2019, 2020), who found more species in adjacent seas. We speculate
that low sampling frequency in the tWP compared with that in the nSCS could be the
main reason of the disagreement. High tintinnid abundance and species richness mainly
appeared at around DCM depths in both the nSCS and tWP (Fig. 5). A high Chl a
environment may be an important factor for influencing tintinnid distribution in oceanic
waters (Dolan and Marrasé, 1995; Suzuki and Taniguchi, 1998).

**5 Conclusions**
The present study reported on diel variations of planktonic ciliate community structure
and relationships with environmental factors in the nSCS and tWP. In the upper 200 m,
the night ciliate abundances and biomasses were higher than in day. Variations in the
weighted mean depth of aloricate ciliates and tintinnids reflected that they did preform
diel vertical migrations in both the nSCS and tWP. Abundance proportions of aloricate-
ciliate large size-fraction and tintinnid species with small lorica oral diameter, exhibited
higher abundances in night than in day, consistently with the night dominance of
smaller cell sizes of food items. Depth and temperature were the main driving factors



for aloricate ciliates, while for several dominant tintinnid species, Chl *a* was another
important driving factor for their diel vertical migration in both the nSCS and tWP.

*Author contributions.* CW and WZ designed the research. CW, YD, MD, LZ, and HL
performed the data analysis. CW, SZ and TX participated the cruises. CW led the
writing of the paper, with input from all co-authors.

*Competing interest.* The authors declare that they have no conflict of interest.

*Disclaimer.* Publisher's note: Copernicus Publications remains neutral with regard to
jurisdictional claims in published maps and institutional affiliations.

*Acknowledgements.* Special thanks to the captains and crews of R.V. "Nanfeng" and
R.V. "Kexue" for their great help in sampling during cruises. We thank Natalie Kim,
PhD, from Liwen Bianji (Edanz) (www.liwenbianji.cn/), for editing the English text of
a draft of this manuscript.

*Financial support.* This work was supported by the China Postdoctoral Science
Foundation (grant number 2020M672149), the Strategic Priority Research Program of
the Chinese Academy of Sciences (No. XDB42000000), the National Natural Science
Foundation of China (grant numbers 41576164), and the Applied Research Project for
Postdoctoral Researchers in Qingdao.

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





**Tables**

**Table 1**. Sampling stations location, sampling time and day/night classification in the northern South China Sea (nSCS) and tropical West Pacific (tWP).

| Seas | stations | Latitude (°N) | Longitude (°E) | Date | Time | Day/Night |
|------|----------|---------------|----------------|------|------|-----------|
| nSCS | TS1 | 19.8531 | 116.1238 | 2017.03.29 | 11:20 | Day |
|      | TS2 | 19.8531 | 116.1238 | 2017.03.29 | 18:42 | Night |
|      | TS3 | 19.8531 | 116.1238 | 2017.03.30 | 0:05 | Night |
|      | TS4 | 19.8531 | 116.1238 | 2017.03.30 | 7:12 | Day |
|      | TS5 | 19.8531 | 116.1238 | 2017.03.30 | 12:20 | Day |
|      | TS6 | 19.8531 | 116.1238 | 2017.03.30 | 18:57 | Night |
|      | TS7 | 19.8531 | 116.1238 | 2017.03.31 | 0:37 | Night |
|      | TS8 | 19.8531 | 116.1238 | 2017.03.31 | 5:11 | Night |
|      | TS9 | 19.8531 | 116.1238 | 2017.03.31 | 11:32 | Day |
| tWP | TS1 | 10.0778 | 140.1889 | 2019.06.02 | 19:35 | Night |
|      | TS2 | 10.0778 | 140.1889 | 2019.06.03 | 0:35 | Night |
|      | TS3 | 10.0778 | 140.1889 | 2019.06.03 | 7:50 | Day |
|      | TS4 | 10.0778 | 140.1889 | 2019.06.03 | 13:30 | Day |
|      | TS5 | 10.0778 | 140.1889 | 2019.06.03 | 19:25 | Night |





**Table 2**. Spearman's rank correlation between the planktonic ciliate (aloricate size-fraction and tintinnid dominant species) abundance (ind. L$^{-1}$) and depth (m), temperature (T, °C), salinity, and chlorophyll $a$ concentrations (Chl $a$, µg L$^{-1}$).

| Seas | Group | Size-fraction/Species | Day | | | | Night | | | |
|---|---|---|---|---|---|---|---|---|---|---|
| | | | Depth | T | Salinity | Chl $a$ | Depth | T | Salinity | Chl $a$ |
| nSCS | Aloricate ciliate | 10-20 µm | -0.728** | 0.655** | 0.558** | 0.312 | -0.748** | 0.763** | 0.570** | 0.305 |
| | | 20-30 µm | -0.820** | 0.761** | 0.574** | 0.159 | -0.827** | 0.836** | 0.573** | 0.134 |
| | | > 30 µm | -0.899** | 0.830** | 0.548** | 0.051 | -0.874** | 0.858** | 0.563** | 0.075 |
| | | All | -0.847** | 0.823** | 0.573** | 0.194 | -0.842** | 0.805** | 0.573** | 0.154 |
| | Tintinnid | *Salpingella faurei* | 0.017 | 0.049 | 0.520** | 0.297 | -0.066 | 0.143 | 0.441** | 0.448** |
| | | *Dadayiella ganymedes* | -0.651** | 0.655** | 0.283 | 0.010 | -0.867** | 0.831** | 0.482** | -0.013 |
| | | *Proplectella perpusilla* | 0.080 | -0.034 | 0.300 | 0.548** | 0.130 | -0.047 | 0.297 | 0.178 |
| | | *Steenstrupiella steenstrupii* | -0.585** | 0.515** | 0.266 | -0.115 | -0.673** | 0.563** | 0.283 | -0.173 |
| | | *Epiplocylis acuminata* | -0.234 | 0.267 | 0.302 | 0.186 | -0.052 | 0.124 | 0.297 | -0.015 |
| | | All | -0.384* | 0.453** | 0.574** | 0.471** | -0.558** | 0.596** | 0.574** | 0.344* |
| | Total | | -0.838** | 0.819** | 0.573** | 0.215 | -0.844** | 0.805** | 0.573** | 0.154 |
| tWP | Aloricate ciliate | 10-20 µm | -0.819** | 0.836** | 0.087 | 0.538* | -0.703** | 0.715** | 0.115 | 0.569** |
| | | 20-30 µm | -0.709** | 0.768** | 0.110 | 0.606** | -0.756** | 0.816** | 0.040 | 0.485* |
| | | > 30 µm | -0.675** | 0.722** | 0.078 | 0.591** | -0.762** | 0.864** | -0.179 | 0.347 |
| | | All | -0.776** | 0.817** | 0.095 | 0.604** | -0.770** | 0.828** | -0.005 | 0.491** |
| | Tintinnid | *S. faurei* | -0.491* | 0.522* | -0.398 | -0.199 | -0.429* | 0.467* | -0.283** | -0.136** |
| | | *P. perpusilla* | -0.387 | 0.388 | -0.078 | 0.020 | -0.107 | 0.114 | 0.040 | 0.406* |
| | | *Ascampbelliella armilla* | -0.653** | 0.732** | -0.483* | -0.161 | -0.547** | 0.629** | -0.148 | 0.280 |
| | | *Acanthostomella minutissima* | -0.181 | 0.116 | 0.249 | 0.572* | -0.195 | 0.220 | 0.420* | 0.759** |
| | | *Eutintinnus hasleae* | -0.228 | 0.376 | 0.000 | 0.431 | -0.199 | 0.181 | 0.043 | 0.057 |
| | | *Canthariella brevis* | -0.841** | 0.800** | -0.544* | -0.272 | -0.768** | 0.776** | -0.409* | -0.195 |
| | | *Metacylis sanyahensis* | -0.364** | 0.420** | -0.088 | -0.044 | -0.279* | 0.333** | -0.002 | 0.337* |
| | | *Protorhabdonella curta* | -0.434 | 0.485* | -0.349 | -0.175 | -0.497** | 0.528** | -0.320 | -0.153 |
| | | All | -0.725** | 0.766** | -0.021 | 0.386 | -0.452* | 0.490** | 0.316 | 0.720** |
| | Total | | -0.778** | 0.820** | 0.080 | 0.583* | -0.747** | 0.804** | 0.040 | 0.537** |

Note: **: $p < 0.01$, *: $p < 0.05$

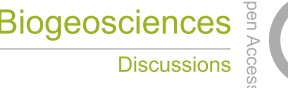

Figures

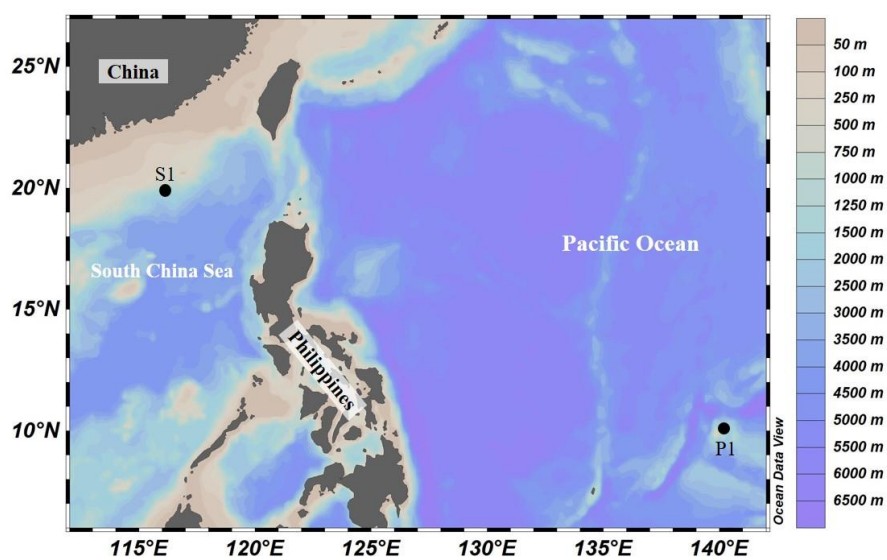

**Figure 1.** Survey stations in the northern South China Sea (nSCS) and tropical West Pacific (tWP).



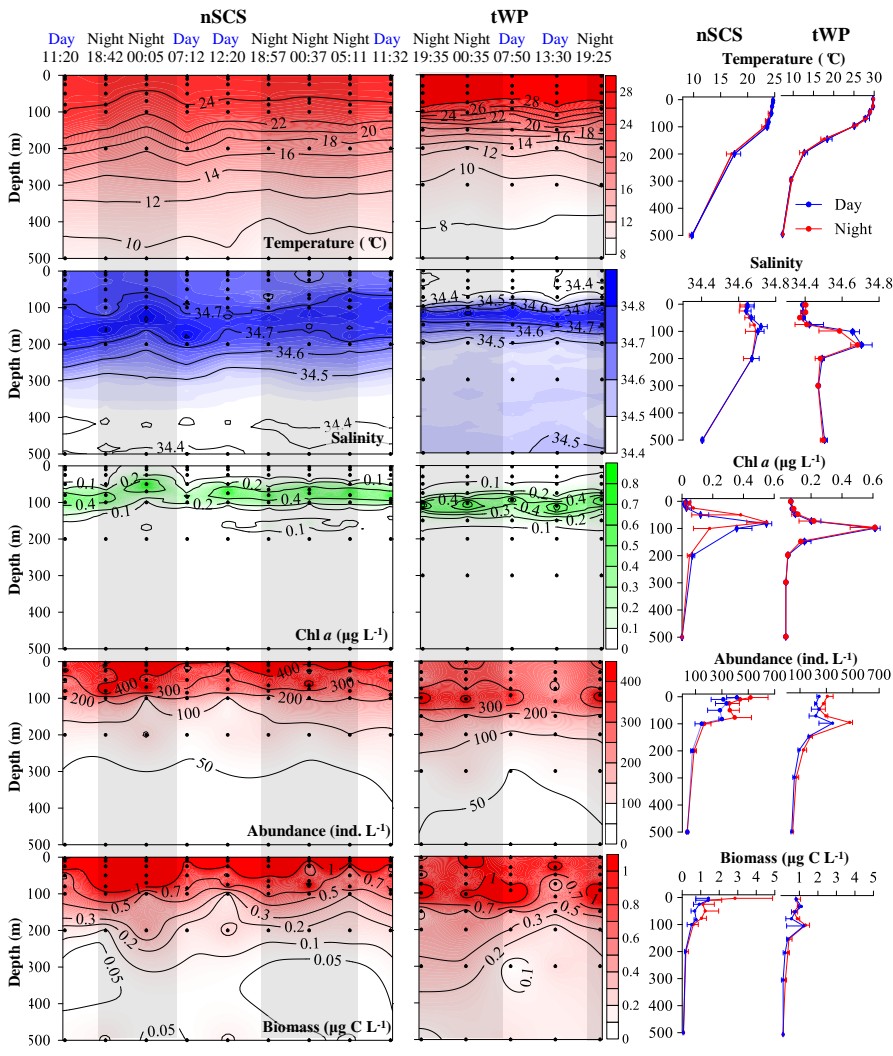

**Figure 2.** Temperature (T), salinity (S), Chlorophyll *a* (Chl *a*), total ciliate abundance and biomass profiles from the surface to 500 m in the nSCS and tWP. Black dots: sampling depths; black shadows: night.



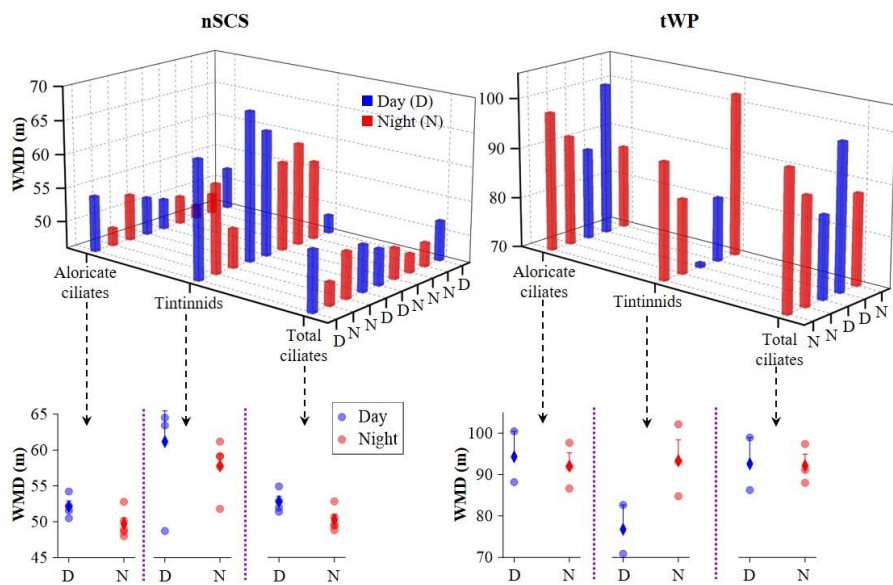

**Figure 3.** Three-dimensional representation for the weighted mean depth (WMD) variations of ciliate abundance during day-night in the nSCS and tWP.



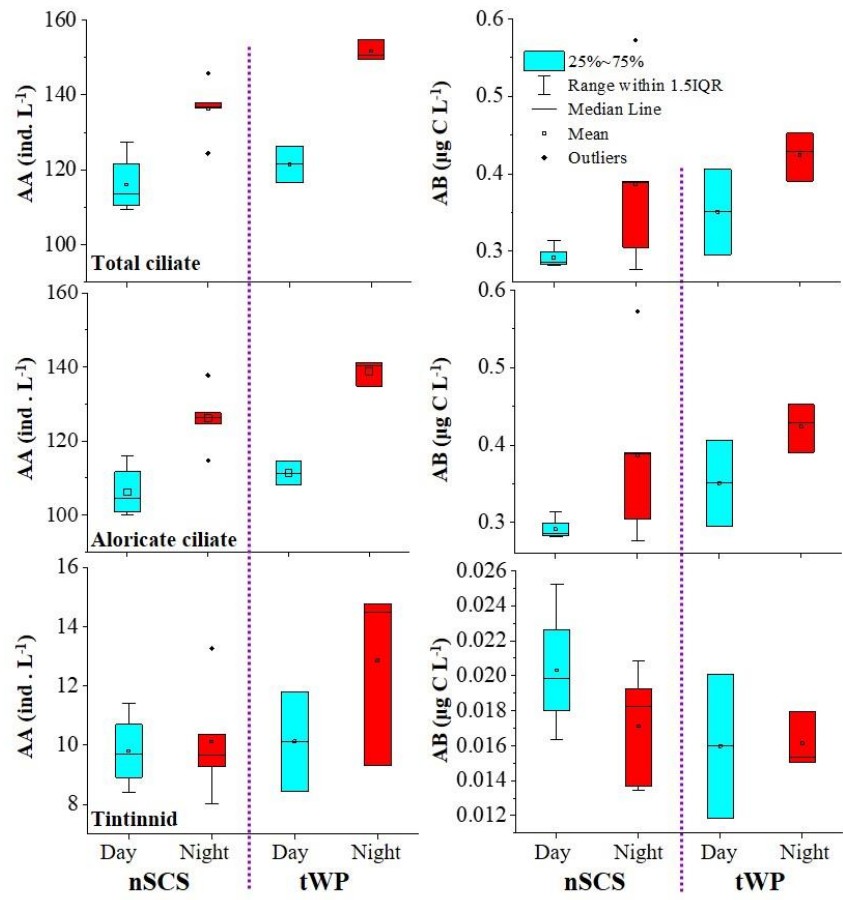

**Figure 4.** Day-night variations of ciliate (total, aloricate ciliate and tintinnid) water column average abundance (AA) and average biomass (AB) in the nSCS and tWP.





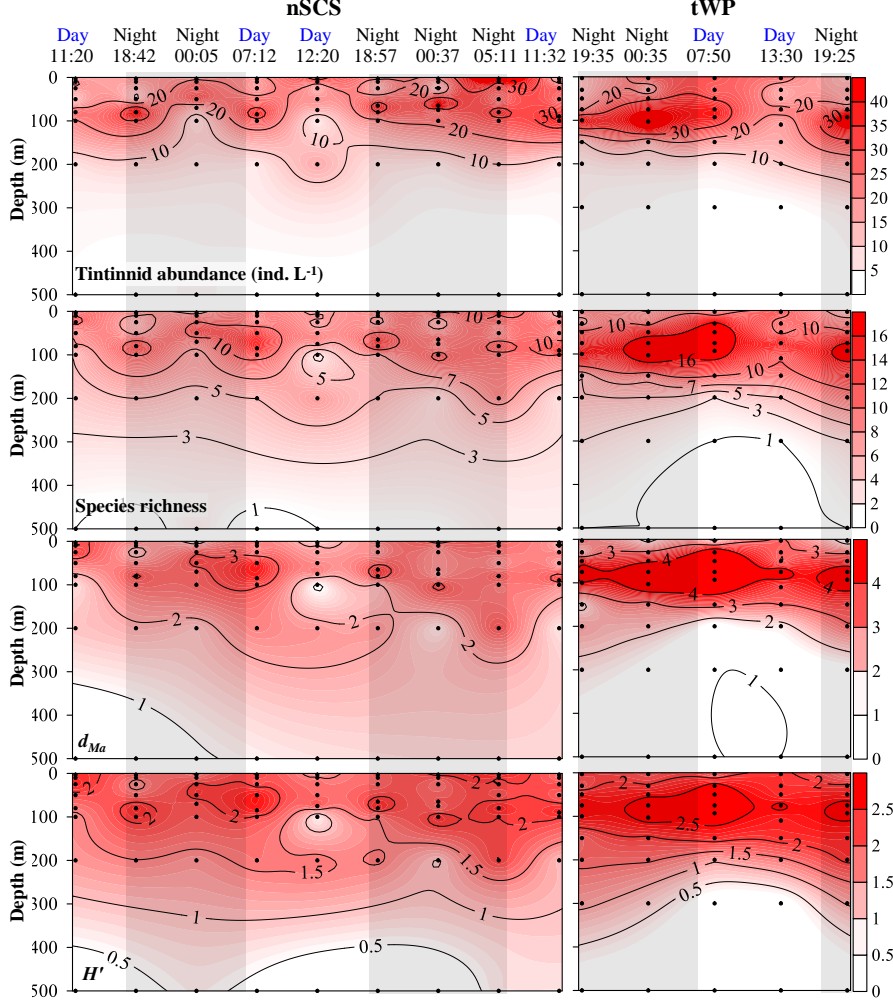

**Figure 5.** Day-night variations of tintinnid abundance, species richness and diversity indices at each layers in the nSCS and tWP. $d_{Ma}$: Margalef index; $H'$: Shannon index. Black shadows: night.



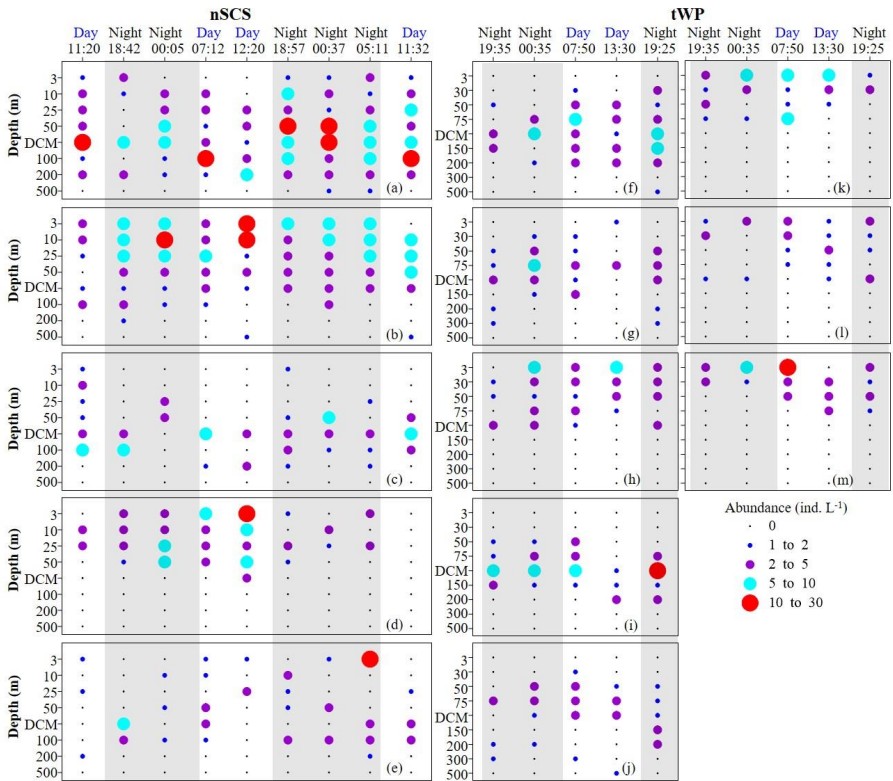

**Figure 6.** Day-night variations of tintinnid dominant species at each layers in the nSCS and tWP. (a) and (f): *Salpingella faurei*; (b): *Dadayiella ganymedes*; (c) and (g): *Proplectella perpusilla*; (d): *Steenstrupiella steenstrupii*; (e): *Epiplocylis acuminata*; (h): *Ascampbelliella armilla*; (i): *Acanthostomella minutissima*; (j): *Eutintinnus hasleae*; (k): *Canthariella brevis*; (l): *Metacylis sanyahensis*; (m): *Protorhabdonella curta*. Black shadows: night.





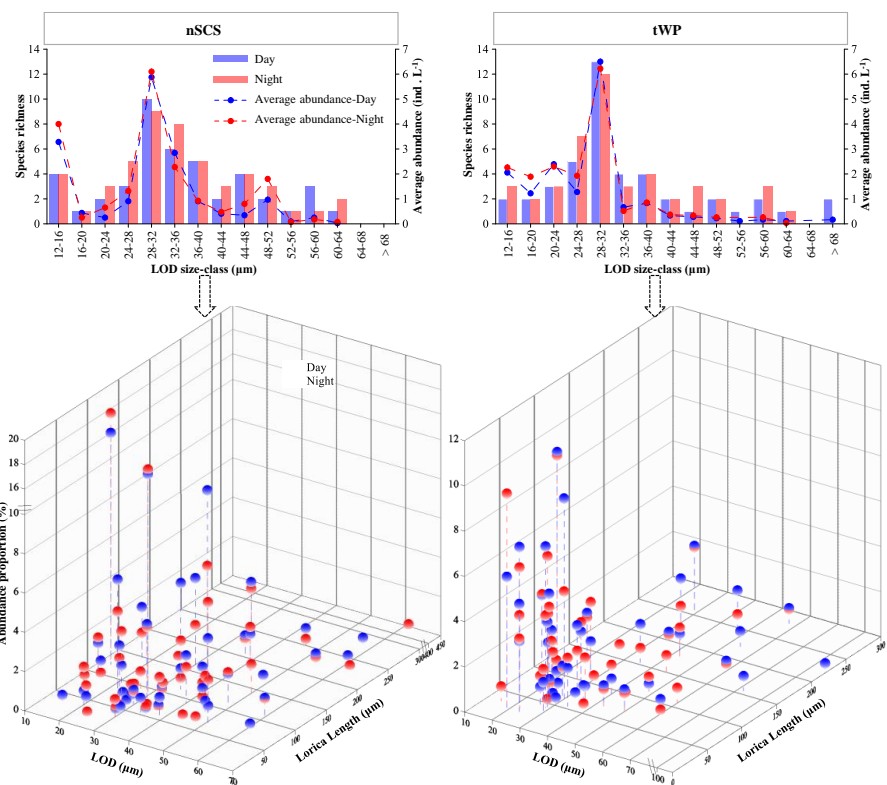

**Figure 7.** Three-dimensional representation for day-night variations of tintinnid species richness, lorica oral diameter (LOD), lorica length, average abundance and abundance proportion in the nSCS and tWP.

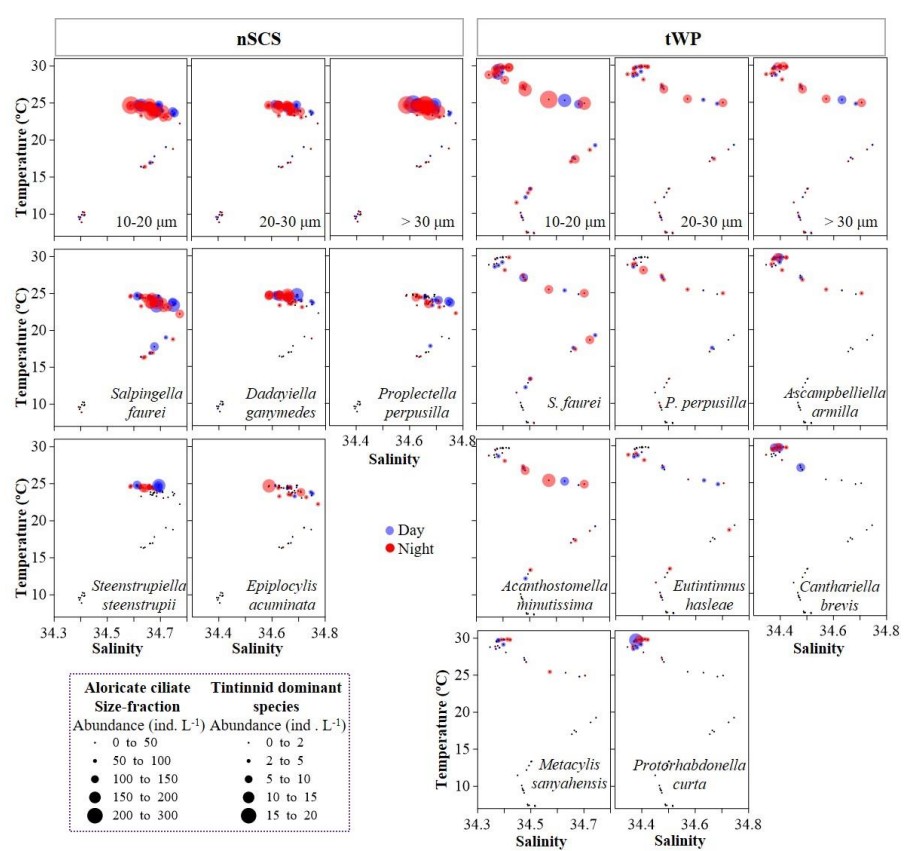


**Figure 8.** Temperature-salinity-plankton diagrams for day-night variations of size-fractions (10-20 μm, 20-30 μm and >30 μm) of aloricate ciliate and tintinnid dominant species in the nSCS and tWP.
