# Peer review of "Diel variations in planktonic ciliate community structure in the northern South China Sea and tropical Western Pacific"

_Biogeosciences, 2022_

## Author Comment (AC1)

Response to reviewers

Dear Editor,

We finished the revision of the manuscript in response to the questions and advices of the two reviewers. The following are the details of our responses to questions and advices of every reviewer. Our revisions were **in the revised manuscript (Manuscript-bg-2022-151-revised clean-R1)**.
The work of the reviewers help improve the quality of the manuscript. We thank the thoughtful advice of the reviewers and hope the revision successfully answered the questions.
Best wishes,

**Wuchang Zhang**
=============================================

**Reviewer 1:**

Wang et al. investigated the diel variations in the abundance, biomass, and community structure of planktonic ciliates in the northern South China Sea and tropical Western Pacific. They found that the abundance and biomass of ciliates at night were slightly higher than in the daytime, and ciliates perform diel vertical migrations. Overall, this study is interesting, and the data is valuable as few studies probed into the diel variations of planktonic ciliates based on time-series data. However, I have several concerns with the results and conclusions. I am afraid that the results presented in this manuscript cannot support their conclusions. Thus, I do not recommend the publication of this manuscript unless they address the following issues.

General comments:
1. My main criticism is that there is no serious statistical testing throughout the manuscript. This is a fatal flaw which undermined the quality of this MS. All results are descriptive and hard to be trusted. The results and conclusion could be different, even opposite, if they conduct statistical testing.
Response: Special thanks for pointing out the fatal flaw of the manuscript. We really appreciated your rigorous and careful checking for our manuscript. As for this problem, we added statistical testing (nonparametric-test, Independent *t*-test, and PERMANOVA analysis) in the manuscript to eliminate the flaw according to your valuable suggestions accordingly **in revised manuscript**. At last, we also added partial Mantel tests to evaluate the correlation between ciliate community and environmental factors.

2. Many descriptions of results are not accurate. The authors seem to describe the results in a general way but ignore some different patterns between day and night and nSCS and tWP. For instance, the authors stated that the abundance and biomass of ciliate were higher at night than in the daytime without any support from statistical testing. In fact, the biomass of ciliates showed no difference between day and night in the upper 200 layers of the tWP based on Fig. 2.
Response: We realized that many descriptions of results are not accurate, and special thanks for pointing out our shortcomings in the manuscript. We accepted suggestions and added statistical

testing (nonparametric-test and Independent *t*-test) in "Results" part accordingly **in revised manuscript**.

As for "the biomass of ciliates showed no difference between day and night in the upper 200 layers of the tWP based on Fig. 2", we revised accordingly **in lines 206-208 in revised manuscript**.

"From surface to 200 m depth, average abundance and biomass of ciliates in the night were higher than in the day in the nSCS. While in the tWP, only average abundance of ciliates at night were higher than at day".

3. One of the main conclusions in this manuscript was that ciliates perform diel vertical migrations. However, the evidence for this conclusion is too weak to support it. The author calculated the weighted mean depth (WMD) of day and night and stated a difference between them. In fact, the difference may not be significant if they conduct statistical testing. According to Fig. 3, we can observe that the average WMD of day and night was similar for aloricate ciliates and total ciliates in the tWP. Based on these results, how can the authors conclude diel vertical migrations for all ciliates? It is interesting to observe the diel vertical migration for ciliates, but more rigorous and robust evidence is highly necessary.

Response: We conducted nonparametric-test to state the significant difference of WMD between day and night and found there were no significant differences between day and night in the nSCS **in lines 218-221 in revised manuscript**. Because of the less data in the tWP, we cannot conduct statistical test to state the significant difference of WMD between day and night.

"The weighted mean depth (WMD) of ciliate community during day and night differed in both the nSCS and tWP, but no significant difference in WMD between day and night for aloricate ciliate (nonparametric-test, $p \geq 0.05$), tintinnid (nonparametric-test, $p \geq 0.05$) and total ciliate (nonparametric-test, $p \geq 0.05$) in the nSCS".

As to the diel vertical migration of ciliate, there were few methods for testing its diel vertical migration in the oceanic seas. The weighted mean depth (WMD) was used to test diel vertical migration in copepods (Frost and Bollens, 1992). To our knowledge, we are the first to use the WMD for testing ciliate diel-vertical-migration. To reduce errors, we treat aloricate ciliate, tintinnid and total ciliate separately. We found that the WMD of aloricate ciliate and tintinnid deferred between day and night in both the nSCS and tWP with migration distance less than 20 m (Fig. 3), supporting our hypothesis that they do perform diel vertical migration. This distance was consistent with the speed of ciliate (1-2.5 m/h) in the eutrophic lakes (Jonsson 1989). We cannot provide more rigorous and robust evidence at this time. But more experiments about diel-vertical-migration by exact ciliate species in culture are put on the agenda in future surveys.

References:

Frost, B. W., and Bollens, S. M.: Variability of diel vertical migration in the marine planktonic copepod Pseudocalanus newmani in relation to its predators, Can. J. Fish. Aquat. Sci., 49, 1137–1141. https://doi.org/10.1139/f92-126, 1992.

Jonsson, P. R.: Vertical distribution of planktonic ciliates–an experimental analysis of swimming behavior, Mar. Ecol. Prog. Ser., 52, 39–53, https://doi.org/10.3354/meps052039, 1989.

4. The manuscript needs to be edited by a native writer. In many places, the text is confusing, and occasionally grammar mistakes are found. One issue is the phrase "in night" should be a grammar mistake. Instead, "in the night" and "at night" are more commonly used. Please check and revise it throughout the manuscript.

Response: We apologized for our grammar mistakes. A professional English editor, Dr. Natalie Kim (from Liwen Bianji), helped us revise the manuscript. We revised "in night" into "at night" **throughout the revised manuscript**.

Specific comments:

5. Line 26 what do you mean by "preformed"? is it "performed"?

Response: We apologized for the mistake here and revised into "performed" **throughout the revised manuscript**.

6. Line 147 change to "The average abundance and biomass of the water column were calculated following……"

Response: We accepted the suggestion and revised accordingly **in lines 149-150 in revised manuscript**.

"The average abundance and biomass of the water column were calculated following Yu et al. (2014) and Wang at al. (2022b)".

7. Lines 150-154 change to "the tintinnid genera are classified into two groups based on Pierce and Turner (1993) and Dolan and Pierce (2013): Cosmopolitan, species ……; Warm Water, species….."

Response: We accepted the suggestion and revised accordingly **in lines 152-156 in revised manuscript**.

"Biogeographically, the tintinnid genera are mainly classified into two groups in the oceanic waters based on Pierce and Turner (1993) and Dolan and Pierce (2013): Cosmopolitan, species distributed widespread in the world ocean; Warm Water, species observed in both coastal systems and open waters throughout the world ocean, but absent from sub-polar and polar waters".

8. Lines 158-159 I don't understand "the midpoint of each sampling layer" is the sampling depth such as 200m? It seems that WMD is an important index for assessing diel vertical migration. It is better to describe it in more detail.

Response: We accepted suggestion and revised this sentence accordingly **in lines 160-161 in revised manuscript**.

"Where $a_i$ is the abundance of ciliate (aloricate ciliate, or tintinnid, or total ciliate) at depth $d_i$, and $d_i$ is the each sampling depth".

9. Lines 192-193 change to "……mainly observed in the upper 100m of nSCS and 150m of tWP, and the values decreased down to 500m depth…"

Response: We accepted the suggestion and revised accordingly **in lines 197-199 in revised manuscript**.

"High ciliate abundance ($\geq$ 200 ind. $L^{-1}$) and biomass ($\geq$ 0.5 µg C $L^{-1}$) values were mainly observed in the upper 100 m of nSCS and the upper 150 m of tWP, and the values decreased down to 500 m depth (Fig. 2)".

10. Lines 194-196 This description seems not accurate. Based on Fig. 2, the highest abundance and biomass occurred in the DCM layers of the tWP. I cannot see a 'peak' for both abundance and biomass in the surface layer of tWP.

Response: In the tWP, average abundance in the surface layer was higher than 30 m depth in both day and night, but not for biomass at day. Thus we revised and rewrote this sentence accordingly **in lines 200-203 in revised manuscript**.

"The vertical profiles of ciliate average abundance showed bimodal (in the surface and DCM layers) patterns throughout day and night in both the nSCS and tWP. But average biomass showed surface-peak in the nSCS and DCM-peak in the tWP (Fig. 2)".

11. Lines 197-198 The average biomass in the upper 200m layer of tWP showed no difference between day and night. I don't think you can draw the conclusion that the average abundance and biomass were higher in the day than at night.

Response: We apologized for the mistake and revised accordingly **in lines 206-208 in revised manuscript**.

"From surface to 200 m depth, average abundance and biomass of ciliates at night were higher than at day in the nSCS. While in the tWP, only average abundance of ciliates at night were higher than at day".

12. Lines 207-209 Were the difference in WMD between day and night significant in both nSCS and tWP? According to Fig. 3, the average WMD of aloricate ciliates and total ciliates in the day and night seems not different in the tWP.

Response: According to nonparametric-test, the difference in WMD between day and night in the nSCS were not significant ($p \geq 0.05$). We revised accordingly **in lines 218-221 in revised manuscript**.

"The weighted mean depth (WMD) of ciliate community during day and night differed in both the nSCS and tWP, but no significant difference in WMD between day and night for aloricate ciliate (nonparametric-test, $p \geq 0.05$), tintinnid (nonparametric-test, $p \geq 0.05$) and total ciliate were (nonparametric-test, $p \geq 0.05$) in the nSCS".

13. Lines 216-217 how to calculate the integrated abundance and biomass? Is it the depth-integrated value? This should be explained clearly in the Method. If the difference is significant, the statistical test result should be presented after this statement.

Response: As for "the integrated abundance and biomass", we made a mistake here and revised into "Average abundance and biomass" **in lines 230-231 in revised manuscript**.

"Average abundance and biomass of ciliate showed different characteristics during day and night in both the nSCS and tWP (Fig. 4)".

As for the statistical test, we used Independent *t*-test to state the significant difference **in lines 231-235 in revised manuscript**.

"In the nSCS, the average water-column abundance of total ciliates (136.3 ± 7.7 ind. L$^{-1}$) (Independent *t*-test, $P < 0.01$) and aloricate ciliates (126.2 ± 8.2 ind. L$^{-1}$) (Independent *t*-test, $P <$

0.01) at night were significant higher than that at day (116.1 ± 8.1 ind. L$^{-1}$, and 106.3 ± 7.3 ind. L$^{-1}$), respectively".

14. Lines 220-223 change this sentence to "the average water-column biomass of tintinnids was lower at night (0.017 ± 0.003 μg C L-1) than in day (0.020 ± 0.004 μg C L-1)."
Response: We accepted the suggestion and revised accordingly **in lines 237-238 in revised manuscript**.
"but the average water-column biomass of tintinnids was lower at night (0.017 ± 0.003 μg C L$^{-1}$) than at day (0.020 ± 0.004 μg C L$^{-1}$)".

15. Line 224 "the water-column average abundances and biomasses" is a bit strange. I think it is the average value of water-column abundance and biomass, as you may first calculate the water-column integrated abundance and biomass and then calculate their mean. Please modify these phrases accordingly.
Response: We accepted the suggestion and modified these phrases accordingly **in lines 238-245 in revised manuscript**.
"In the tWP, the average water-column abundance and biomass of total ciliates, aloricate ciliates and tintinnids at night were higher than at day (Fig. 4). As to variations between two seas, the average water-column abundance and biomass of total ciliates and aloricate ciliates at night and day were higher in the tWP than that in the nSCS (Fig. 4), but not significant (Independent t-test, P ≥ 0.05). Although average water-column abundance of tintinnids in both the night and day in the tWP were higher than that in the nSCS, their average water-column biomass were lower in the tWP than in the nSCS (Fig. 4)".

16. Line 224 This sentence described the difference between tWP and nSCS but started with "As to night and day variations"? Please rephrase it.
Response: We revised accordingly **in line 240 in revised manuscript**.
"As to variations between two seas,……".

17. Lines 258-260 It is better to state that (1) the species prefer 50m and DCM; (2) the high abundance occurred at night more frequently. Please rephrase this sentence to make it clear.
Response: We accepted the suggestion and revised this paragraph accordingly **in line 275-286 in revised manuscript**.
"As for dominant species in the nSCS, *S. faurei* and *Epiplocylis acuminata* prefer 50 m and DCM layers and their high abundance occurred at night more frequently. In contrast, in the surface layer, *Dadayiella ganymedes* and *Steenstrupiella steenstrupii* were present in higher abundance at day than at night (Fig. 6; Figs. S3 and S4). The *P. perpusilla* prefer 25 m and 50 m layers at night. But at DCM and 100 m layers, its high abundance occurred at day more frequently (Fig. 6; Figs. S3 and S4). In the tWP, *S. faurei*, *P. perpusilla*, *Ascampbelliella armilla*, *Acanthostomella minutissima* and *Metacylis sanyahensis* prefer DCM layers and their high abundance occurred at night more frequently. While for *Canthariella brevis* and *Protorhabdonella curta*, their abundance were higher at day than at night in surface layers. With regard to *Eutintinnus hasleae*, its abundance was higher at night than at day in waters ranged from 50 to 200 m (except DCM) (Fig. 6; Figs. S3 and S4)".

18. Lines 274-276 This sentence contradicts what you just said. What do the numbers in brackets mean?

Response: We revised this sentence accordingly **in lines 289-293 in revised manuscript**. The numbers in brackets mean species richness of each seas.

"High species richness and average abundance in tintinnid LOD (lorica oral diameter) size-classes had some differences throughout the day and night in both the nSCS and tWP (Fig. 7). Species richness at night over the nSCS (49 species) and tWP (45 species) were slightly higher than at day (nSCS: 44 species, tWP: 44 species), respectively (Fig. 7; Table S2)".

19. Line 278 What do you mean by "Between day and night"? for both day and night?

Response: We revised this sentence accordingly **in line 295 in revised manuscript**.

"For both day and night, ……".

20. Line 280 add "in the nSCS and tWP," before respectively.

Response: We accepted the suggestion and revised accordingly **in lines 295-298 in revised manuscript**.

"For both day and night, the second highest species richness in the nSCS and tWP were 32-36 μm and 24-28 μm LOD size-class, respectively, while the second highest average abundance were 12-16 μm and 20-24 μm LOD size-class in the nSCS and tWP, respectively".

21. Line 281 Again, it contradicts the first sentence.

Response: We revised the first sentence accordingly **in lines 289-291 in revised manuscript**.

"High species richness and average abundance in tintinnid LOD (lorica oral diameter) size-classes had some differences throughout the day and night in both the nSCS and tWP (Fig. 7)".

22. Lines 283-291 What is the size of the dominant species such as S. faurei and D. ganymedes? Why go back to species? Is there species information in Fig. 7?

Response: The size of tintinnid species included lorica length and lorica oral diameter. In Fig. 7 of the species abundance proportion, each dot means each species abundance proportion variations in the day (blue dots) and night (red dots). We also revised the Fig. 7 accordingly **in revised Fig.7 in revised manuscript**.

23. Line 297 what do "temperature and salinity range" mean? Is it quantitative? Please specify how to get the conclusion about the broader range.

Response: We deleted this sentence and revised the following sentence accordingly **in revised manuscript**.

24. Lines 300-302 do you mean the range in the nSCS is smaller than that in the tWP? The temperature variation in the tWP is larger than nSCS. It is not surprising that you observed this result.

Response: We revised this sentence accordingly **in lines 315-318 in revised manuscript**.

"Regarding differences between the two sites, the average temperature of each aloricate ciliate size-fraction with abundance > 100 ind. $L^{-1}$ in the nSCS (23.1-24.8 ℃, average 24.3 ±0.5 ℃) was lower than that in the tWP (24.8-29.8 ℃, average 27.8 ±1.9 ℃) (Fig. S5)".

25. Line 324 change "phototrophic" to "mixotrophic".

Response: We accepted suggestion and revised accordingly **in lines 346-347 in revised manuscript**. "excluding the mixotrophic ciliate *Mesodinium rubrum,* which exhibits obvious diel vertical migration".

26. Line 330-331 How do previous studies access ciliate diel vertical migration?

Response: At first, there were few studies about ciliate diel vertical migration. Most of them studied ciliate diel vertical migration in shallow waters of coastal seas or lakes. Previous studies access ciliate diel vertical migration through the following steps: 1, selected one or several stations, and took water samples in every 2 h (Rossberg and Wickham, 2008), or 3 h (Gu et al., 2022), or 4 h (Stoecker et al., 1989; Pérez et al., 2000), 6 h (Suzuki and Taniguchi, 1997) in one or several days; 2, counted ciliate (ciliate assemblage or specific species) abundance at each depth; 3, calculated their variations between day and night, and eventually assessed ciliate diel vertical migration. Our methods were same with previous studies.

References:

Gu, B. W., Huang, H., Zhang, Y. Z., Li, R., Wang, L., Wang, Y., Sun, J., Wang, J. N., Zhang, R., Jiao, N. Z., and Xu, D. P.: High dynamics of ciliate community revealed via short-term, high-frequency sampling in a subtropical estuarine ecosystem. Front. Microbiol., 13, 797638, https://doi.org/10.3389/fmicb.2022.797638, 2022.

Pérez, M. T., Dolan, J. R., Vidussi, F., and Fukai, E.: Diel vertical distribution of planktonic ciliates within the surface layer of the NW Mediterrean (May 1995), Deep Sea Res. I., 47, 479–503, https://doi.org/10.1016/S0967-0637(99)00099-0, 2000.

Rossberg, M., and Wickham, S. A.: Ciliate vertical distribution and diel vertical migration in a eutrophic lake, Fund. Appl. Limnol., 171, 1–14, https://doi.org/10.1127/1863-9135/2008/0171-0001, 2008.

Stoecker, D. K., Taniguchi, A., and Michaels, A. E.: Abundance of autotrophic, mixotrophic and heterotrophic ciliates in shelf and slope waters, Mar. Ecol. Prog. Ser., 50, 241–254, https://doi.org/10.3354/meps050241, 1989.

Suzuki, T., and Taniguchi, A.: Temporal change of clustered distribution of planktonic ciliates in Toyama Bay in summers of 1989 and 1990, J. Oceanogr., 53, 35–40, https://doi.org/10.1007/BF02700747, 1997.

27. Line 369 what does "biomass decreased" mean? Please state it more clearly.

Response: We accepted suggestion and revised accordingly **in lines 404-406 in revised manuscript**. "Our results showed that tintinnid abundance was higher but biomass was lower at night than at day in both the nSCS and tWP (Figs. 4 and 7)".

28. Line 378-379 Tintinnids only feed on picoplankton and bacteria? I think they also feed on nanoplankton. Also, in the oligotrophic ocean, ciliates mainly feed on phytoplankton rather than bacteria, while ciliates in freshwater largely feed on bacterial (Weisse and Montagnes 2021; DOI: 10.1111/jeu.12879).

Response: We revised this sentence accordingly **in lines 414-417 in revised manuscript**.

"As for heterotrophic microzooplankton tintinnids, photosynthetic organisms, e.g., nanoplankton (nanoflagellates), are important food items for influencing their abundance and composition in the oligotrophic seas (Pitta et al., 2001; Weisse and Montagnes, 2022)".

29. Line 388 "bimodal type" is not accurate.
Response: We revised this sentence accordingly **in lines 427-429 in revised manuscript**.
"Abundance peaks of planktonic ciliates occurred in surface and DCM layers in both the nSCS and tWP, but highest abundances occurred in surface layer of the nSCS, and DCM layer of the tWP (Fig. 2)".

30. Line 391 change "who" to "which".
Response: We accepted this suggestion and revised accordingly **in lines 429-430 in revised manuscript**.
"Our results are consistent with Wang et al. (2019), which discovered this phenomenon and proposed a hypothesis to verify it".

31. Line 413-414 change to "the ciliate abundance and biomass at night were higher than that in the day".
Response: We accepted this suggestion and revised accordingly **in lines 459-460 in revised manuscript**.
"In the upper 200 m, the ciliate abundance and biomass at night were higher than that at day".

32. Line 415 what does "preform" mean? A typo?
Response: We apologized for this mistake. This word was a typo problem. We revised accordingly **in lines 461-462 in revised manuscript**.
"…they did perform diel vertical migrations in both the nSCS and tWP".

33. Line 419 In fact, depth and temperature are strongly correlated. The prey availability (Chla) may also affect the distribution of aloricate ciliates.
Response: We admit that prey availability (Chl $a$) may had important influence on the distribution of aloricate ciliates. But through both Spearman's rank correlation and Mantel test between aloricate ciliate with environmental variables, we found that aloricate ciliate had no significant correlation with Chl $a$ in both day and night in the nSCS (Tables 2 and 3). Thus we did not list Chl $a$ as one of the main driving factors for aloricate ciliates.

34. Fig. 3 Do the diamonds indicate the average values of WMD? Figure legend is needed for illustrating the symbols in detail
Response: The diamonds indicate the average values of WMD in Fig. 3. We added figure legend for illustrating the symbols in detail and revised accordingly **in revised Fig. 3 in revised manuscript**.

**Reviewer 2:**

The authors try to understand the diel variation of planktonic ciliates in northern South China Sea and tropical West Pacific. This investigation requires tremendous efforts and times. It must be very

valuable information for understanding of plankton ecology and ciliates community. This manuscript, however, have weak point on (1) sampling design and (2) reliability of data obtained for short period.

1. In your study, sampling location is fixed. You do not trace the same ciliate community conveyed by the continuous water flow. Small-scaled patchiness and small-scaled non-homogenous distribution of planktonic organisms are frequently observed in any sea areas. You had better to explain the validity of your sampling design with referring to this point.

Response: Our sampling locations were fixed and we could not trace the same ciliate community conveyed by the continuous water flow. We also recognized that small-scaled patchiness and small-scaled non-homogenous distribution of planktonic organisms are frequently observed in any sea areas. But all above items had less connection with our sampling designs. In the manuscript, we aimed to find out diel variations in planktonic ciliate community structure in the nSCS and tWP. We treated aloricate ciliates, tintinnids, and total ciliates separately. Through WMD, we wanted to recognize whether each of them perform diel vertical migration in the nSCS and tWP. During initial sampling design stage of this manuscript, we found that both the nSCS and tWP belonged to oceanic seas, and they had some similarities and differences of ciliate diel vertical distributions. Thus we put ciliate diel vertical variations together in both the nSCS and tWP.

2. Sampling period at tWP station might be too short. Daytime sampling was carried out only two times in the same day. This sampling design might be unreasonable for comparing ciliate community structure between daytime and nighttime. You should explain the reliability of your results obtained under this sampling design.

Response: We admit that sampling period at tWP station is too short. But we have no selection and opportunity to carry out a long period survey. The chance for the survey about ciliate diel variations in the tropical western Pacific Ocean is extremely precious. We also revised accordingly **in lines 84-87 in revised manuscript**.

"Despite their important role in marine microbial food webs, our knowledge of ciliate assemblage diel variations in tropical oceanic waters are limited due to their inaccessibility for oceanographic surveys".

In 'Results'

3. Sentence-to-sentence connection is frequently not smooth; meaning of each chapter is incomprehensible. This might be mainly due to the excessive change of subject in each sentence. You had better to rewrite sentences in each chapter of result part.

Response: We appreciated your valuable suggestions and rewrote sentences in each chapter of result part throughout all "Results" part **in revised manuscript**.

4. Statistical test (significant analysis in statistics) should be indispensable when you compare distributional depth of ciliates or compare standing crops of ciliates between daytime and nighttime or between nSCS and tWP.

Response: We accepted suggestion and added statistical test (Independent $t$-test, nonparametric-test, PERMANOVA analysis) to compare distributional depth of ciliates or compare standing crops of

ciliates between daytime and nighttime in the nSCS **in revised manuscript.** Because of less data in the tWP, we cannot perform Independent *t*-test or nonparametric-test to testing the difference.

5. Fig. 7 (three-dimensional graphs). Data points are many and complicated. You had better to improve these graphs more comprehensible.
Response: Information of Fig. 7 included tintinnid lorica length (*x*-axis), lorica oral diameter (*y*-axis) and each species abundance proportion (*z*-axis). As for the species abundance proportion in Fig. 7, each dot means variations of each species abundance proportion at day (blue dots) and night (red dots). We also revised the Fig. 7 to improve these graphs more comprehensible **in revised Fig. 7 in revised manuscript**.

6. Fig. 8, Fig. S4, Fig. 5S
Data points are many and sometimes overlapped. I can not understand the difference between daytime pattern and nighttime pattern.
Response: We put data of day and night together in order to better describing their difference in Fig. 8, thus many data points were overlapped. For purpose of better visualization, we increased the transparency of each data **in revised Fig. 8 in revised manuscript**. We also accepted suggestion and revised Figs. S4 and S5 accordingly **in revised Figs. S5, S6 and S7 in Supplement-R1**.

In 'Discussion'
7. Line 331, 'diel-vertical-migration'
You had better to discuss some reasonable factors affecting this 'diel vertical migration'.
You had better to check that migrating scale (or distance) of ciliates estimated from the diel variation of WMD is comparable to the reported swimming speed (or swimming ability) of marine planktonic ciliates.
Response: As for reasonable factors affecting this 'diel vertical migration', we accepted this suggestion and revised accordingly **in lines 361-371 in revised manuscript**.
"There were multiple factors that could influence diel-vertical-migration behavior of ciliate (food items concentration and quality, predator avoidance, light intensity, body metabolic rates, etc) in various seas (e.g., McLaren 1974; Loose et al., 1993; Rossberg and Wickham, 2008). In the oligotrophic seas, the phytoplankton assemblage was dominated by Prochlorococcus, Synechococcus and picoeukaryotes, and they showed different diel variations (e.g., Vaulot and Marie, 1999; Oubelkheir and Sciandra, 2008). As important food items of ciliate, heterotrophic bacteria also displayed clear daily oscillations in the oligotrophic Ionian Sea (Mediterranean) (Oubelkheir and Sciandra, 2008). Thus we speculate that diel variation of food items was possibly the main reason in determining ciliate diel-vertical-migration behavior in the oligotrophic tropical seas".

As for migrating scale (or distance) of ciliates, we referred some references and revised accordingly **in lines 349-360 in revised manuscript**.
"In contrast, other studies provided evidence that ciliates indeed perform diel vertical migration (Dale 1987; Jonsson 1989; Pérez et al., 2000; Rossberg and Wickham, 2008). e.g., in the eutrophic lakes with steep light gradients, ciliate have been observed to move vertically at speeds of 1–2.5 m/h (Jonsson 1989). The weighted mean depth (WMD) was used to test diel vertical migration in

copepods (Frost and Bollens, 1992). To our knowledge, we are the first to use the WMD for testing ciliate diel-vertical-migration. We found that the WMD of aloricate ciliates and tintinnids differed between day and night in both the nSCS and tWP with migration distance less than 20 m (Fig. 3), supporting our hypothesis that they do perform diel vertical migration, and this distance was consistent with the speed of ciliate (1-2.5 m/h) in the eutrophic lakes (Jonsson 1989) and the northwestern Mediterranean Sea (migrate distance: 20-30 m) (Pérez et al., 2000)".

8. Line 335, 'The ciliate vertical distribution patterns were the same between day and night'
This sentence must be in contradiction with the sentence of 'WMD of aloricate ciliates and tintinnids diferred between day and night (line 332)'. It must be also in conflict with the sentence of 'ciliate abundance and biomass were higher in night than in day (line 342)'.
Response: We revised this paragraph accordingly **in lines 372-384 in revised manuscript**.
"The ciliate abundance was high in surface and DCM layers in both day and night of both the nSCS and tWP. These results were similar to previous ones established in the western Pacific Ocean (Yang et al., 2004; Sohrin et al., 2010; Wang et al., 2019, 2020, 2021b). However, the studies that previously investigated the ciliate vertical distribution, did not assess potential differences between day and night in vertical direction. Therefore, our study provides more accurate data on ciliate diel-vertical-migration in the nSCS and tWP. Additionally, our results in the upper 200 m provide evidence that ciliate abundance were higher at night than at day in both the nSCS and tWP (Fig. 2). Zooplankton distribution in waters mainly depends on phytoplankton presence (Daro 1988; Ursella et al., 2018). Thus, it is possible that the availability of more food items (flagellates, picoeukaryotes, *Prochlorococcus*, *Synechococcus* and heterotrophic bacteria) at night than at day explains the higher ciliate abundance at night (Olli 1999; Oubelkheir and Sciandra, 2008; Li et al., 2022)".

9. Line 378, 'Heterotrophic tintinnids feed on ------ in the ocean.'
I can not understand the meaning of connection between this sentence and the previous sentence. This discontinuity might make the chapter unclear. You had better to rewrite the sentence more clearly.
Response: We revised this sentence accordingly **in lines 414-422 in revised manuscript**.
"As for heterotrophic microzooplankton tintinnids, photosynthetic organisms, e.g., nanoplankton (nanoflagellates), are important food items influencing their abundance and composition in the oligotrophic seas (Pitta et al., 2001; Weisse and Montagnes, 2022). Our study showed that tintinnid abundance at night was higher than at day for two possible reasons: 1) oceanic tintinnid species have stronger cell division in midnight than at day in tropical Pacific waters (Heinbokel 1987); and 2) predation on picoplankton, nanoplankton and heterotrophic bacteria occurred primarily at night (Tsai et al., 2005; Ribalet et al., 2015; Connell et al., 2020)".

10. Line 394, 'nutrient loaded'
You had better to descript theoretical processes from this nutrient load to ciliate increase. In Fig. 2, chl-a concentration in the surface layer was very low and phytoplankton must not be promoted by the loaded nutrients.
Response: We accepted suggestion and revised this sentence accordingly **in lines 434-439 in revised manuscript**.

"Nutrients are material basis for the growth of microphytoplankton and heterotrophic bacteria. High nutrient concentrations always accompanied with high abundance of microphytoplankton and heterotrophic bacteria in surface waters in the oligotrophic tropical seas, which further affected and determined microzooplankton abundance and composition (Caron, 1994; Kirchman, 1994; Song, 2011; Zhang et al., 2016; Ma et al., 2020)".

11. Line 404, 'We speculate that ----- could be the main reason of the disagreement.'
You had better to descript the theoretical reason on your speculation.
Response: We accepted suggestion and revised this sentence accordingly **in lines 447-452 in revised manuscript**.
"Low sampling frequency is often accompanied by low species richness (Dolan et al., 2007, 2009). The total samples in the tWP (45 samples) and nSCS (72 samples) were much lower than in previous studies (≥ 100 samples) (Li et al., 2018; Wang et al., 2019, 2020). Thus we speculate that low sampling frequency in our results could be the main reason for the disagreement".